# Cholesterol-Lowering Activity of Vitisin A Is Mediated by Inhibiting Cholesterol Biosynthesis and Enhancing LDL Uptake in HepG2 Cells

**DOI:** 10.3390/ijms24043301

**Published:** 2023-02-07

**Authors:** Yangbing Yuan, Yuanqin Zhu, Yawen Li, Xusheng Li, Rui Jiao, Weibin Bai

**Affiliations:** Department of Food Science and Engineering, Institute of Food Safety and Nutrition, Jinan University, Guangzhou 510632, China

**Keywords:** pyranoanthocyanins, Vitisin A, cholesterol biosynthesis, HMGCR, LDL uptake

## Abstract

Pyranoanthocyanins have been reported to possess better chemical stability and bioactivities than monomeric anthocyanins in some aspects. The hypocholesterolemic activity of pyranoanthocyanins is unclear. In view of this, this study was conducted to compare the cholesterol-lowering activities of Vitisin A with the anthocyanin counterpart Cyanidin-3-*O*-glucoside(C3G) in HepG2 cells and to investigate the interaction of Vitisin A with the expression of genes and proteins associated with cholesterol metabolism. HepG2 cells were incubated with 40 μM cholesterol and 4 μM 25-hydroxycholeterol with various concentrations of Vitisin A or C3G for 24 h. It was found that Vitisin A decreased the cholesterol levels at the concentrations of 100 μM and 200 μM with a dose–response relationship, while C3G exhibited no significant effect on cellular cholesterol. Furthermore, Vitisin A could down-regulate 3-hydroxy-3-methyl-glutaryl coenzyme A reductase (HMGCR) to inhibit cholesterol biosynthesis through a sterol regulatory element-binding protein 2 (SREBP2)-dependent mechanism, and up-regulate low-density lipoprotein receptor (LDLR) and blunt the secretion of proprotein convertase subtilisin/kexin type 9 (PCSK9) protein to promote intracellular LDL uptake without LDLR degradation. In conclusion, Vitisin A demonstrated hypocholesterolemic activity, by inhibiting cholesterol biosynthesis and enhancing LDL uptake in HepG2 cells.

## 1. Introduction

Coronary heart disease (CHD) is one of the leading causes of death in the world [1]. Lowering blood total cholesterol and low-density lipoprotein (LDL) cholesterol are effective strategies in reducing overall CHD risk [2]. The Mediterranean diet, rich in plant-based foods and monounsaturated fat, is an increasingly popular diet pattern that is widely accepted for its cardioprotective effects [3,4]. “The French paradox” is also well known, stating that lower CHD incidence was partially associated with a daily intake of red wine in moderation [5]. Numerous studies have described that moderate consumption of alcoholic beverages reduced the risk of developing cardiovascular diseases (CVD) and elucidated the benefit of red wine over other alcoholic beverages [6,7]. Meta-analysis and experimental studies have mainly attributed this outcome to the presence of polyphenolic compounds such as resveratrol, anthocyanin, catechin, epicatechin, and quercetin in red wine [8,9].

Anthocyanins were identified as the main pigments that contributed to most of the overall red wine color; their concentrations are reported to reach up to 2000 mg/L in red wine [10]. Anthocyanins are considered to effectively prevent CHD mainly because of their lipid-lowering properties including suppressed cholesterol synthesis [11,12], stimulated cholesterol efflux [13,14], increased lipolysis [15], decreased lipogenesis [16], and so on [17].

In recent years, a new class of anthocyanin-derivatives called pyranoanthocyanins, which are formed from anthocyanins and ethanol metabolites that appear in wine during processing and aging, have received great attention [18]. According to their structural differences, pyranoanthocyanins are divided into different groups (Figure 1), including Vitisins-, Methyl-, Phenolic-, Flavanol-, Portisins-, Oxovitisins-, and Dipolymer-pyranoanthocyanins [19]. Most pyranoanthocyanins present a higher color intensity, better chemical stability, and higher antioxidant potential and other bioactivities compared to their anthocyanin counterparts in vitro [20,21,22]. However, the cholesterol-lowering effect of pyranoanthocyanins and its underlying mechanism is still unclear.

Cholesterol metabolism is controlled by many different genes and proteins in the liver and intestine. The liver is the main site of cholesterol biosynthesis and regulation. Sterol regulatory element-binding protein 2 (SREBP2) is a transcriptional factor of cholesterol biosynthesis enzymes such as 3-hydroxy-3-methyl-glutaryl coenzyme A reductase (HMGCR), the only rate-limiting enzyme of cholesterol endogenous synthesis [23]. The liver also administrates cholesterol uptake from the bloodstream through the low-density lipoprotein receptor (LDLR) or scavenger receptor class B type 1 (SR-B1) [24]. LDLR plays a key role in capturing LDL cholesterol through the vesicle endocytosis mechanism, and the circulation of LDLR from lysosomes to cell membranes could be inhibited by proprotein convertase subtilisin/kexin type 9 (PCSK9) by promoting LDLR degradation [25]. However, SR-B1 mainly mediates cholesterol uptake from high-density lipoproteins (HDL) [26]. Furthermore, cholesterol efflux was mediated by the ATP-binding cassette (ABC) transporter superfamily to reduce excessive cholesterol accumulation, including ABC subfamily A member 1 (ABCA1) and ABC subfamily G (ABCG) members 5 and 8. ABCA1 is crucial in transporting cholesterol into plasma, combining with lipid-poor apolipoprotein A-I (apoA-I) to generate nascent HDL [27]. ABCG5 and ABCG8 were abundantly expressed on the apical surface of enterocytes and the membrane of hepatocytes, forming a heterodimer to transport cholesterol and further extract by bile salts [28]. Hepatocytes also secrete cholesterol into the intestinal lumen or bile duct via bile acids [29]. The classical pathway of bile acid synthesis is administrated by cholesterol-7α hydroxylase (CYP7A1), and overexpression of CYP7A1 could expand the hydrophobic bile acid pool for fecal cholesterol excretion [30]. Extensive research has affirmed that nuclear receptors play important roles in the response to excess cholesterol, such as peroxisome proliferator-activated receptors γ (PPARγ), which could increase the expression of ABCA1 and ABCG5/8 to facilitate cholesterol efflux and cholesterol excretion [31]. Liver X receptor α (LXRα) also controls cholesterol synthesis, cholesterol excretion, and bile acid synthesis in multiple ways, and ABCA1, ABCG5/8, and CYP7A1 were identified as direct target genes of LXRα in mammals [32,33].

Vitisin A is the main pyranoanthocyanin detected in aged wines, which forms upon the reaction between the enol form of pyruvic acid and the anthocyanins [34,35] (Figure 2). The present study aims to (i) compare the cholesterol-lowering activities of Vitisin A with its anthocyanin counterpart Cyanidin-3-*O*-glucoside (C3G) in human hepatocellular carcinoma (HepG2) cells, and (ii) investigate how Vitisin A interacts with the expression of genes and proteins associated with cholesterol metabolism.

## 2. Results

### 2.1. Preparation of C3G and Vitisin A

Vitisin A was synthesized from the reaction of black soybean peel extract anthocyanins with pyruvic acid, and purified with MPLC. The purified compounds were obtained and identified with HPLC (Figure 2A,B). The purity of Vitisin A and C3G calculated through the area normalization method were 94.8% and 95%, respectively. Consistent with Fulcrand’s results, the LC/MS chromatogram showed that the maximum absorption wavelength of Vitisin A is 501 nm, and *m/z* 355 was the main fragment ion of Vitisin A, which formed from *m/z* 517 losing a glucoside [35] (Figure 2C). 

### 2.2. Cellular Toxicity of Anthocyanins and Atorvastatin in HepG2 Cells

HepG2 cells were exposed to different concentrations of C3G and Vitisin A (50, 100, and 200 μM) and atorvastatin (10, 20, 40 μM) for 24 h, and the cell viability curve was drawn in Figure 3 based on the CCK-8 method. The cells were photographed using an inverted microscope to observe the morphological changes. It was found that C3G or Vitisin A treatments ranging from 50 to 200 μM and Atorvastatin (Ator) ranging from 10 to 40 μM had no cytotoxicity to HepG2 cells (Figure 3A–C). Cells of all treatment groups also presented regular proliferation and growth status compared with the control group (Figure 3D). Hence, the data showed that both C3G and Vitisin A ranging between 50 and 200 μM could be used for further experiments, as 10 μM Ator was used as a positive control.

### 2.3. Vitisin A Decreased Intracellular Cholesterol in HepG2 Cells

HepG2 cells were incubated for 24 h with 40 μM cholesterol, 4 μM 25-hydroxycholeterol, and Vitisin A at different concentrations. As a positive control, 10 μM atorvastatin was used. Total cellular cholesterol (TC) was measured with a TC kit. The results showed that the cholesterol-lowering effect of Vitisin A has a dose–response relationship; 100 μM Vitisin A lowered cholesterol concentration by 10% and 200 μM Vitisin A significantly decreased cholesterol by 30% (Figure 4A). The cholesterol-lowering effect of Vitisin A was also investigated in contrast to C3G., The cells were treated with 200 μM Vitisin A, 200 μM C3G, or 10 μM Ator together with 40 μM cholesterol and 4 μM 25-hydroxycholesterol for 24 h (Figure 4B). Specifically, while 200 μM Vitisin A effectively decreased the cholesterol level by 33% comparative to the model group, 200 μM C3G had no significant effect on cellular cholesterol.

### 2.4. Vitisin A Regulated the Gene Expressions of Cholesterol Metabolism

To further investigate the mechanisms through which Vitisin A decreased cellular cholesterol, the gene expressions involved in cholesterol metabolism were analyzed with real-time PCR. HepG2 cells were incubated with 40 μM cholesterol, 4 μM 25-hydroxycholesterol, and different concentrations of Vitisin A (50, 100, and 200 μM) for 24 h. As shown in Figure 5B, Vitisin A (50, 100, and 200 μM) markedly down-regulated HMGCR mRNA expression approximately 0.69-fold as compared with the high-cholesterol model group (*p* < 0.01). Next, the mRNA expression of LDLR was enhanced consistently with the increased concentration of Vitisin A. Vitisin A treatment also up-regulated the mRNA expressions of SREBP2, PPARγ, and ABCA1, but the influence of Vitisin A on PCSK9, LXRα, and ABCG8 was insignificant (Figure 5).

### 2.5. Vitisin A Regulated the Protein Expressions of Cholesterol Metabolism

Western blot was performed to observe the effects of Vitisin A on protein expressions in cholesterol metabolism. The protein levels of SREBP2, HMGCR, PCSK9, LXRα, ABCA1, and ABCG5 were down-regulated by Vitisin A, while the protein levels of PPARγ and CYP7A1 had no significant changes (Figure 6). It is worth noting that the expression levels of HMGCR and PCSK9 were significantly decreased in the 200 μM Vitisin A-treated group, while the expression of LDLR simultaneously increased in comparison with the model group. Therefore, Vitisin A was available to reduce PCSK9 expression and maintain the LDLR level, which was beneficial for LDLR circulation and LDL cholesterol elimination.

## 3. Discussion

Anthocyanins, the main source of color in red wines, are also widely found in the red–blue color of abundant fruits and vegetables. There may be a causal relationship between consumption of abundant anthocyanins in red wine and prohibition of heart attacks, and decreased blood total cholesterol and LDL cholesterol level [36]. The cholesterol-lowering effect of anthocyanins have been thoroughly investigated by many studies [37,38]. While pyranoanthocyanins, a family of anthocyanin derivatives formed during wine aging, have been investigated contributing to the progressive shift of the red–purple color of young wines to a more orange color in aged wines, there has been no comparative study on the cholesterol-lowering activity of pyranoanthocyanins and anthocyanins. The present study was the first to investigate the hypocholesterolemic activity of Vitisin A (the most representative pyranoanthocyanin) and its underlying mechanisms in HepG2 cells. The results showed that Vitisin A was more effective in decreasing cholesterol level in HepG2 than its anthocyanin counterpart C3G, and the underlying mechanisms of cholesterol-lowering activity of Vitisin A were associated with (i) down-regulating HMGCR to inhibit cholesterol biosynthesis through an SREBP2-dependent mechanism and (ii) up-regulating LDLR and blunting the secretion of PCSK9 protein to promote intracellular LDL uptake without LDLR degradation.

Prominently, Vitisin A could decrease HMGCR expression and increase LDLR expression at both the mRNA and protein levels to inhibit cholesterol biosynthesis and enhance cholesterol uptake, which were governed by SREBP2 at a transcriptional level. In this study, HepG2 cells were administrated with 40 μM cholesterol and 4 μM 25-hydroxycholeterol to elevate the intracellular cholesterol level, which led to a reduction in mRNA levels of SREBP2, HMGCR, LDLR, and PCSK9. Thus, extra cholesterol in the cell had a negative impact on cholesterol biosynthesis and uptake, which is in agreement with previous studies [39,40]. Vitisin A could effectively down-regulate HMGCR and up-regulate LDLR expression, especially in the high-dose group. Some in vivo studies revealed that blueberry anthocyanins, black bean husk and black rice anthocyanin extracts were also able to down-regulate the gene expression of hepatic HMGCR, thereby inhibiting the synthesis of cholesterol in the liver [41,42,43]. An anthocyanin-rich fraction has been proven to markedly enhance LDL cholesterol uptake by increasing the LDLR protein level [12,44]. As pyranoanthocyanins are derivatives of anthocyanins, their regulations of cholesterol biosynthesis and uptake may be similar to those of anthocyanins. Moreover, it is known that statins could enhance PCSK9 expression to induce LDLR degradation, thus reducing the LDL cholesterol response to statin therapy [45]. It is noteworthy that, in the present study, Vitisin A clearly suppressed the protein expression of PCSK9, which enhanced LDL cholesterol uptake without PCSK9-induced degradation of LDLR.

The present study also demonstrated that Vitisin A up-regulated the transcriptional levels of PPARγ and ABCA1, but down-regulated the protein levels of ABCA1 and LXRα. ABCA1 is a key factor in cholesterol homeostasis, as it mediates the secretion of cellular-free cholesterol to an extracellular acceptor, apolipoprotein A-1, to form HDL [46], which is the first step of reverse cholesterol transport (RCT). Previous studies have demonstrated that PPARγ and LXR ligands cooperatively regulate ABCA1 and cholesterol efflux in both macrophages and hepatocytes [45,46]. The regulation of Vitisin A on ABCA1 expression was also mediated by both PPARγ and LXR. Furthermore, the cellular ABCA1 protein level was regulated by transcriptional activation and protein degradation [47]. The turnover of ABCA1 protein was rapid, with a short half life of less than 1 h. Thus, it was speculated that Vitisin A decreased ABCA1 protein levels by promoting its protein degradation. Similar results were also observed in the treatment of HepG2 cells with unsaturated fatty acids and PPARγ agonist [48,49]. Furthermore, Vitisin A had no positive effect on ABCG5/8 and CYP7A1 mediated by LXRα, whose protein expression has been shown to be down-regulated by the negative feedback mechanism.

## 4. Materials and Methods

### 4.1. Preparation of C3G and Vitisin A

C3G and Vitisin A were prepared in the current research with purities more than 95%. C3G was purified from black soybean peel extract anthocyanins (Baichuan, Xi’an, China) according to our previous study [50]. For Vitisin A, 10 mg/mL black soybean peel extract anthocyanins were blended with 364 mg/mL pyruvate acid (Macklin, Shanghai, China) in flasks (pH = 3.0), kept in the dark and shaken in a 30 °C water bath for 15 days. After that, the residue was concentrated with rotary evaporation and purified by the method of medium-pressure liquid chromatography (MPLC) described by our lab [51]. A Shimadzu UFLC-MS-8045 system (Shimadzu, Kyoto, Japan) coupled with an Ultimate 3000 HPLC-DAD system (Thermo Scientific, Dreieich, Germany) was utilized for the identification and quantification of C3G and Vitisin [52]. Briefly, water containing 2% formic acid and acetonitrile were mobile phases A and B, respectively. The flow rate was 0.3 mL/min and the column temperature was set at 30 °C. The chromatographic gradient was: 6% of B at injection, increased linearly to 13% B over the first 6 min, then to 30% B in 20 min, then to 95% B in 2 min, and maintained for 2 min, then back to the initial condition in 2 min and equilibrated for 4 min. LC-MS comprised an ion trap mass spectrometer with an electron spray ionization source (ESI) set to positive ionization mode with nitrogen used as atomization gas and dry gas, flow rate 3 L/min and 10 L/min; probe temperature, 300 °C; DL temperature, 250 °C; convert dynode, 10 kV; detector voltage, 2.42 kV; scan range, *m/z* 100–1000.

### 4.2. Cell Culture and Treatments

Human hepatocellular carcinoma cells (HepG2) were purchased from the Cell Library of the Chinese Academy of Sciences. Cells were cultured in DMEM supplemented with 10% FBS, 1% of amphoteric enzyme B (100 U/mL), penicillin (100 μg/mL), and streptomycin mixture (Gibco/Invitroge, Carlsbad, CA, USA). The cells were preserved at 37 °C with 5% CO_2_ at constant humidity. In our previous study, cells were incubated with 40 μM cholesterol and 4 μM 25-hydroxycholesterol (Sigma, St. Louis, MO, USA) to elevate cellular cholesterol [53]. C3G and Vitisin A were dissolved in DMSO at specified concentrations (0, 50, 100, 200 mM) as stock solutions, which was diluted by complete DMEM to the final concentration of DMSO at 0.1%.

### 4.3. Measurement of Anthocyanins and Atorvastatin Effect on Cellular Toxicity and Morphology

A Cell Counting Kit-8 (CCK-8) (Beyotime, Beijing, China) was used in the measurement of anthocyanins’ cellular toxicity. HepG2 cells in the logarithmic phase were seeded on a 96-well plate (1 × 10^4^ cells per well) and incubated for 12 h. Later, discarding the old medium from the plates, 100 μL fresh medium containing gradient concentrations of C3G and Vitisin A (50, 100, 200 μM) were added to each group. After 24 h, the old medium was replaced with CCK-8 solution (diluting with DMEM in 1:9) and incubated at 37 °C for 1 h. The optical density (OD) was measured with an Infinite^®^ F50 ELISA reader (Tecan, Zurich, Switzerland) at 450 nm. Similarly, HepG2 cells were seeded in 6-well plates (5 × 10^5^) for 12 h and exposed to 0, 50, 100, or 200 μM C3G or Vitisin A for 24 h, then the morphological effects were photographed and analyzed using an inverted microscope (CNOPTEC, Chongqing, China). The cytotoxic effect of Ator ranging from 10 to 40 μM was also examined in HepG2 cells.

### 4.4. Measurement of Total Cholesterol

HepG2 cells were seeded at a density of 5 × 10^5^ cells/well onto a 6-well plate. After 12 h of adherence, the medium was removed, and the cells were washed twice with PBS. 40 μM cholesterol and 4 μM 25-hydroxycholesterol with different concentrations (50, 100, 200 μM) of C3G, Vitisin A, and 10 μM Ator were added to the well to incubate for 24 h. Ator (Macklin, Shanghai, China) was used as a positive control. Then, cells were washed twice with PBS, 100 μL lysis buffer was added, and the lysate was collected to centrifuge at 13,000× *g* for 10 min. The supernatant was used to determine the total cholesterol (TC) by the enzymatic method according to the total cholesterol kit (Applygen, Beijing, China).

### 4.5. Real-Time PCR

Total RNA was extracted from HepG2 cells following the instructions of the HiPure Total RNA Mini Kit (Meiji, Guangzhou, China) and transcribed into complementary DNA (cDNA) by the PrimeScript RT reagent Kit (Accurate, Changsha, China). Quantitative real-time polymerase chain reaction (qRT-PCR) was performed with the SYBR Premix Ex Taq^TM^ II Kit (Accurate, Changsha, China), targeting SREBP2, HMGCR, LDLR, PCSK9, PPARγ, LXRα, ABCA1, and ABCG5. Primer sequences of all target genes and *β*-actin were synthesized from Sangon (Shanghai, China) and shown in Table 1. The mRNA levels were normalized to *β*-actin. Forward primers and reverse primers were synthesized by Sangon Biotech Co., Ltd. (Shanghai, China)

### 4.6. Western Blot

Total protein was obtained from HepG2 cells according to the method previously described in our lab [54]. In short, cells were lysed with lysis buffer (radio immunoprecipitation assay: Phenylmethanesulfonyl fluoride = 100:1) (Beyotime, Beijing, China) on ice and centrifuged at 13,000× *g* for 10 min. The supernatant was used for total protein quantifications using a BCA kit (Applygen, Beijing, China). The proteins with equal concentrations (40 μg) were separated on 8% sodium dodecyl sulfate-polyacrylamide gel (SDS-PAGE) and transferred to polyvinylidene difluoride membranes for electrophoresis, with some modification of the electrophoresis conditions: molecular mass < 130 kDa in 300 mA, 1 h; high molecular mass (>130 kDa) in 300 mA, 1.5 h. Afterwards, membranes were incubated at 4 °C overnight with the described primary antibodies. Antibodies for ABCA1, ABCG5, and CYP7A1 were obtained from Absin (Shanghai, China). The antibody for SREBP2 was purchased from Santa Cruz (Shanghai, China). Antibodies for PPARγ, LXRα, PCSK9, LDLR, HMGCR, and *β*-actin were provided by Proteintech (Wuhan, China). After incubation with HRP secondary anti-mouse antibody (Proteintech, Wuhan, China) or HRP secondary anti-rabbit antibody (Thermo, Chantilly, VA, USA) for 1 h, bands were stained with ECL reagent and detected using a Clinx Chemi 5300Pro (Shanghai, China). Grayscale intensities were analyzed using Clinx Image Analysis and presented as fold changes of *β*-actin.

### 4.7. Data Analysis

Data were analyzed with Graph Pad Prism (8.0), all presented in mean ± standard deviation (mean± SD). One-way analysis of variance (ANOVA) was used for statistical analysis between groups. ^#^
*p* < 0.05, ^##^
*p* < 0.01, ^###^
*p* < 0.001, * *p* < 0.05, ** *p* < 0.01, *** *p* < 0.001, and **** *p* < 0.0001 indicate a statistically significant difference.

## 5. Conclusions

In conclusion, Vitisin A was more efficient in decreasing cholesterol level compared with C3G in HepG2 cells. The cholesterol-lowering activity of Vitisin A was mediated with inhibited cholesterol biosynthesis by down-regulating HMGCR, as well as enhancing LDL cholesterol uptake by up-regulating LDLR and down-regulating PCSK9, which were governed by SREBP2. These results suggest that Vitisin A may create a new accessible therapeutic intervention for hypercholesterolemia (Figure 7).

## Figures and Tables

**Figure 1 ijms-24-03301-f001:**
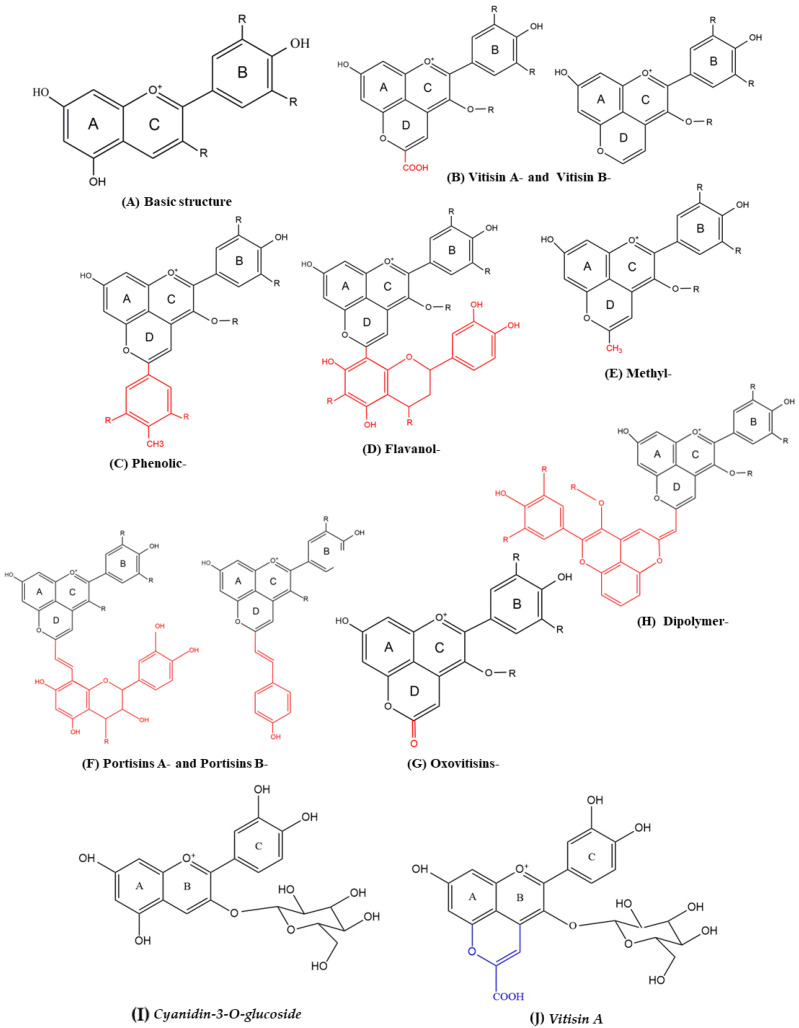
Several common pyranoanthocyanins’ structures. (**A**) Basic structure of anthocyanins. Structures of Vitisin A- or Vitisin B- (**B**), Phenolic- (**C**), Flavanol- (**D**), Methyl- (**E**), Portisin A- or Portisins B- (**F**), Oxovitisins- (**G**), and Dipolymer- (**H**) pyrananthocyanins. The structure of Cyanidin-3-*O*-glucoside (**I**) and Vitisin A (**J**) used in the current research.

**Figure 2 ijms-24-03301-f002:**
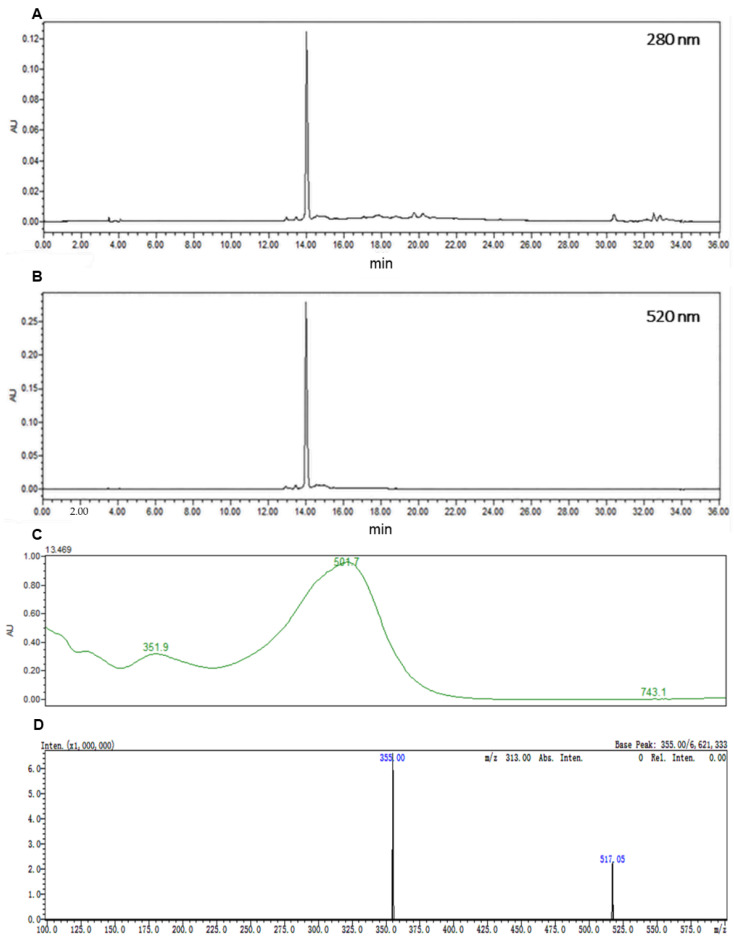
Identification and quantification of Vitisin A. HPLC-DAD profiles of Vitisin A at 280 nm (**A**) and 520 nm (**B**). (**C**) Vitisin A characteristic absorption curve. (**D**) The LC-MS spectrogram of Vitisin A.

**Figure 3 ijms-24-03301-f003:**
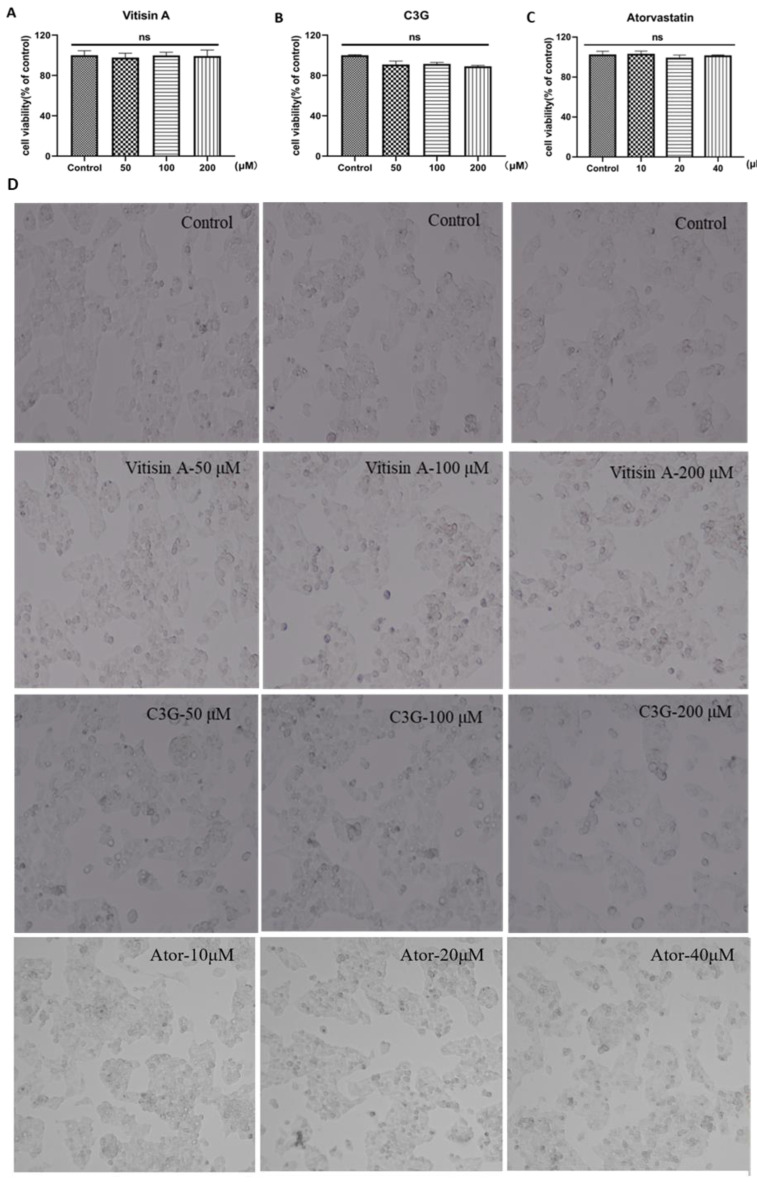
Effects of C3G, Vitisin A and Ator on the viability of HepG2 cells. (**A**,**B**) Cells were cultured with 50, 100, and 200 μM of C3G and Vitisin A for 24 h. (**C**) Cells were cultured with 10, 20, and 40 μM of Ator for 24 h. The cell viability of control was set at 100%. (One-way ANOVA in comparison between all groups, mean ± SD. ns, no significance) (**D**) Cells’ morphological changes with anthocyanins and Ator treatment (100×).

**Figure 4 ijms-24-03301-f004:**
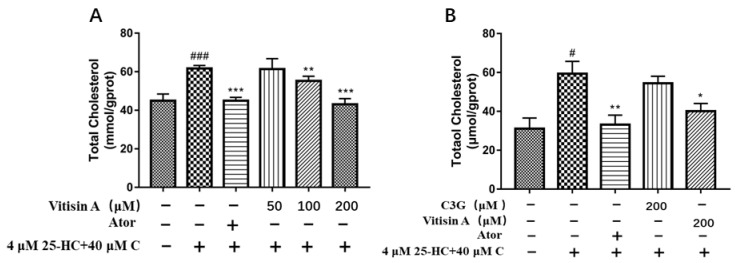
Effect of anthocyanins on the total cholesterol level in HepG2 cells. (**A**) Effects of 200 μM C3G and Vitisin A on TC in HepG2 cells. Cells were incubated with 40 μM cholesterol and 4 μM 25-HC, and 200 μM C3G, Vitisin A or 10 μM Ator for 24 h. (**B**) Effect of 0, 50, 100, and 200 μM Vitisin A on TC in HepG2 Cells. Cells were incubated with 40 μM cholesterol + 4 μM 25-HC and multiple concentrations of Vitisin A (50, 100 and 200 μM) and 10 μM Ator for 24 h. (One-way ANOVA in comparison between all groups. The results are expressed as mean ± SD, ^#^
*p* < 0.05, and ^###^
*p* < 0.001, compared with the blank group. * *p* < 0.05 and ** *p* < 0.01, *** *p* < 0.001, compared with the model group.).

**Figure 5 ijms-24-03301-f005:**
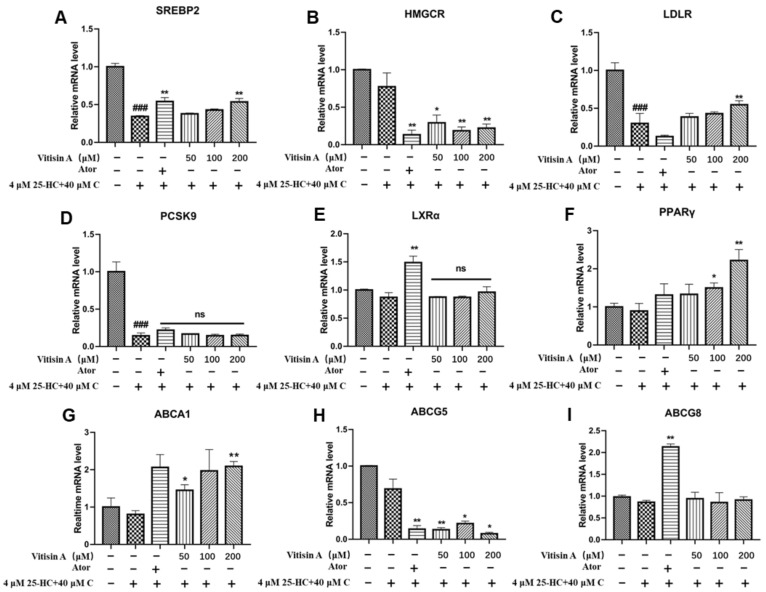
Effects of Vitisin A on mRNA expression of cholesterol metabolism in HepG2 cells. Cells were incubated with 40 μM cholesterol, 4 μM 25-HC, and different concentrations of Vitisin A (50, 100, and 200 μM) or 10 μM Ator for 24 h. The mRNA expression of SREBP2 (**A**), HMGCR (**B**), LDLR (**C**), PCSK9 (**D**), LXRα (**E**), PPARγ (**F**), ABCA1 (**G**), ABCG5 (**H**), and ABCG8 (**I**). (One-way ANOVA in comparison between all groups. The results are expressed as mean ± SD, ^###^
*p* < 0.001, compared with the blank group. * *p* < 0.05 and ** *p* < 0.01, compared with the model group. ns, no significance).

**Figure 6 ijms-24-03301-f006:**
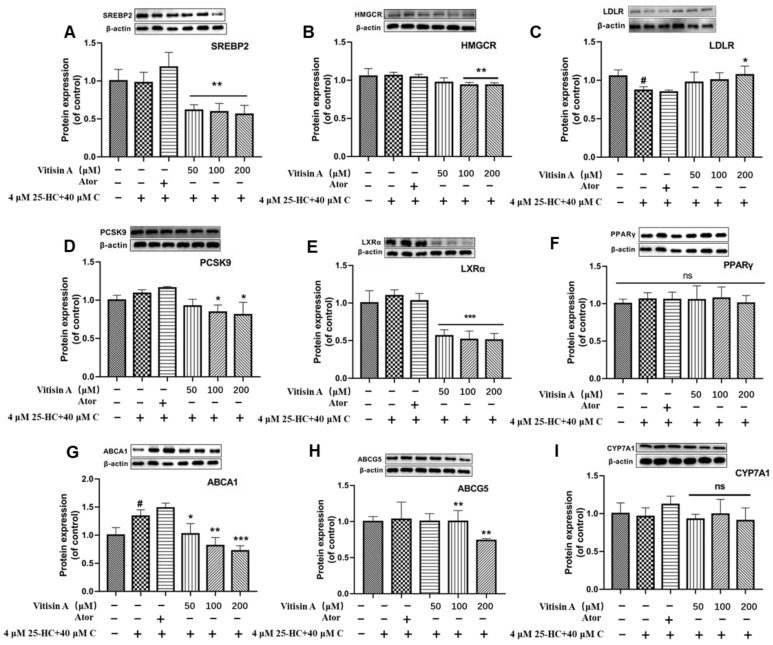
Effects of Vitisin A on protein expressions of cholesterol metabolism in HepG2 cells. Cells were incubated with 40 μM cholesterol, 4 μM 25-HC, and different concentrations of Vitisin A (50, 100, and 200 μM) and 10 μM Ator for 24 h. Representative photographs and grayscale analysis of proteins involved in SREBP2 (**A**), HMGCR (**B**), LDLR (**C**), PCSK9 (**D**), LXRα (**E**), PPARγ (**F**), ABCA1 (**G)**, ABCG5 (**H**), and CYP7A1 (**I**). (One-way ANOVA in comparison between all groups. The results are expressed as mean ± SD, ^#^
*p* < 0.05, compared with the blank group. * *p* < 0.05 and ** *p* < 0.01, *** *p* < 0.001, compared with the model group. ns, no significance).

**Figure 7 ijms-24-03301-f007:**
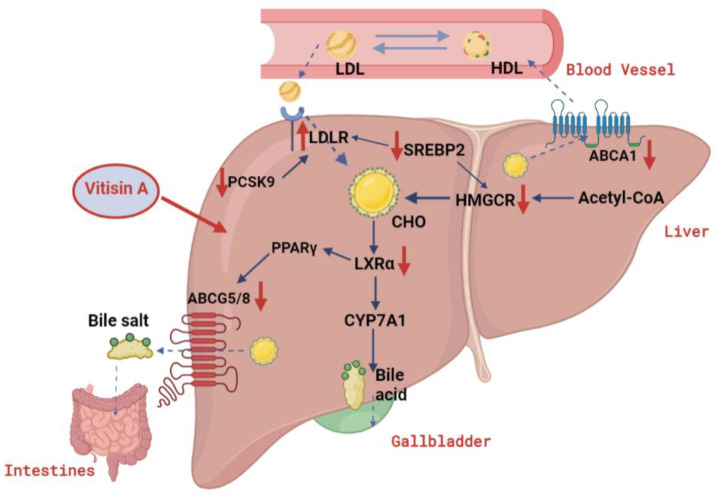
A hypothetical mechanism of Vitisin A in cholesterol metabolism.

**Table 1 ijms-24-03301-t001:** The information of primers.

Gene	Accession No.	Primer Sequence (5′→3′)	
*Srebp2*	NM_004599	Forward	AACGGTCATTCACCCAGGTC
		Reverse	GGCTGAAGAATAGGAGTTGCC
*Hmgcr*	NM_001130996	Forward	CTTGTGTGTCCTTGGTATTAGAGCTT
		Reverse	GCTGAGCTGCCAAATTGGA
*Ldlr*	NM_001195800	Forward	CAAAGTCTGCAACATGGCTAGAGA
		Reverse	GTTGTCCAAGCATTCGTTGGTC
*Pcsk9*	NM_174936	Forward	CCTGGAGCGGATTACCCCT
		Reverse	CTGTATGCTGGTGTCTAGGAGA
*Pparγ*	NM_138711	Forward	GGGATCAGCTCCGTGGATCT
		Reverse	TGCACTTTGGTACTCTTGAAGTT
*Lxrα*	NM_001251934	Forward	CCTTCAGAACCCACAGAGATCC
		Reverse	ACGCTGCATAGCTCGTTCC
*Abca1*	NM_005502	Forward	ACCCACCCTATGAACAACATGA
		Reverse	GAGTCGGGTAACGGAAACAGG
*Abcg5*	NM_022436	Forward	AGCAAGGAACGGGAAATAGA
		Reverse	CAGGAGAACACCCAGTTTAGAG
*β-actin*	NM_001101	Forward	TGGCACCCAGCACAATGAA
		Reverse	CTAAGTCATAGTCCGCCTAGAAGCA

## Data Availability

Data is contained within the article.

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
