# Peer review of "Cholesterol-Lowering Activity of Vitisin A Is Mediated by Inhibiting Cholesterol Biosynthesis and Enhancing LDL Uptake in HepG2 Cells"

_ijms, 2023, doi:10.3390/ijms24043301_

Round 1

Reviewer 1 Report

Yuan and co-authors aimed to compare the cholesterol-lowering activities of Vitisin A with its anthocyanin counterpart Cyanidin-3-O-gluco-side(C3G) in HepG2 cells and to investigate how Vitisin A interacts with the expression of genes and proteins associated with cholesterol metabolism. It is a interesting work. But there are major issue should be focused.

1. Figure 1,  why there are red and blue?  Two H?

2. Figure 2, the vertical and horizontal coordinates are not clear?

3. Why the authors use described dose of Vitisin A?

4. Why the authors use the HepG2 cells not the Mice?

Reviewer 2 Report

In this paper the authors investigated the Cholesterol-lowering activity of Vitisin A on HepG2 cells.

This is interesting and important topic; however, I believe that there are flaws in the experiments, analysis and interpretation of the data that do not permit publication of the work in its current form. It requires major revision

It is noted that the manuscript needs careful editing  (grammar, spelling, and sentence structure).

In addition to considering a grammatical review, the authors can improve the discussion of this interesting manuscript. Comments have been included in the draft of the manuscript.

Round 2

Reviewer 1 Report

The revised manuscript should be published.

Reviewer 2 Report

The authors made all the recommended changes.

I would consider the manuscript for publication in its present form.